# Comprehensive detection of analytes in large chromatographic datasets by coupling factor analysis with a decision tree

Sungwoo Kim[1], Brian M. Lerner[2], Donna T. Sueper[2], Gabriel Isaacman-VanWertz[1]

[1]Charles E. Via Jr. Department of Civil and Environmental Engineering, Virginia Tech, Blacksburg, VA, 24061, USA
[2]Aerodyne Research, Inc., Billerica, MA, 01821, USA

*Correspondence to*: Gabriel Isaacman-VanWertz (ivw@vt.edu)

**Abstract.** Environmental samples typically contain hundreds or thousands of unique organic compounds, and even minor components may provide valuable insight into their sources and transformations. To understand atmospheric processes, individual components are frequently identified and quantified using gas chromatography/mass spectrometry. However, due

to the complexity and frequently variable nature of such data, data reduction is a significant bottleneck in analysis. Consequently, only a subset of known analytes is often reported for a dataset, and a large amount of potentially useful data are discarded. We present here an automated approach of cataloging and potentially identifying all analytes in a large chromatographic dataset and demonstrate the utility of our approach in an analysis of ambient aerosols. We use a coupled factor analysis/decision tree approach to de-convolute peaks and comprehensively catalog nearly all analytes in a dataset.

Positive Matrix Factorization (PMF) of small sub-sections of multiple chromatograms is applied to extract factors that represent chromatographic profiles and mass spectra of potential analytes, in which peaks are detected. A decision tree based on peak parameters (e.g., location, width and height), relative ratios of those parameters, peak shape, noise, retention time, and mass spectrum is applied to discard erroneous peaks and combine peaks determined to represent the same analyte.  With our approach, all analytes within the small section of the chromatogram are cataloged, and the process is repeated for overlapping

sections across the chromatogram, generating a complete list of the retention times and estimated mass spectra of all peaks in a dataset. We validate this approach using samples of known compounds and demonstrate the separation of poorly resolved peaks with similar mass spectra and the resolution of peaks that appear in only a fraction of chromatograms. As a case study, this method is applied to a complex real-world dataset of the composition of atmospheric particles, in which more than 1100 unique chromatographic peaks are resolved, and the corresponding peak information along with their mass spectra are

cataloged.

## 1 Introduction

Atmospheric samples are highly complex and often contain multiple thousands of compounds (Goldstein and Galbally, 2007) with a potentially wide range of physicochemical properties and multiple isomers. Valuable information relating to the sources and chemistry of atmospheric components can be extracted from these compounds, however the complexity of the samples

requires analytical techniques to effectively separate those compounds. Gas chromatography (GC), when combined with mass spectrometry (MS) as a detection method, is one of the most widely used analytical methods in chemical analysis due to high sensitivity, low limits of detection, and high chemical resolution (Hübschmann, 2015). Though used frequently for analysis of atmospheric samples, the complex nature of atmospheric data yields substantial challenges, in particular co-elution of many chromatographic peaks. In some cases, co-elution can be so complex that resolution and integration of individual components cannot be readily achieved, and data is treated as an "unresolved complex mixture" (Zhang et al., 2014). The resolution of GC can be expanded by coupling multiple columns in series, and comprehensive two-dimensional gas chromatography (GC×GC) can provide greater sensitivity and resolution of complex mixtures (Bertsch, 1999; Phillips and Beens, 1999). This technique has yielded valuable insights into atmospheric composition (Hamilton, 2010), but the increased complexity of the instrumentation and more stringent requirements for the mass spectrometer (e.g., time resolution faster than ~50 Hz (Worton et al., 2012)) has limited adoption of GC×GC. Furthermore, despite the higher resolving power, co-elution of peaks still occurs (Potgieter et al., 2016) when highly complex samples are analyzed, and challenges remain in the data analysis. Therefore, it is consequently common for analyses of environmental data to focus on the resolution and quantification of only a subset of specific analytes of interest and leave a large fraction of data unprocessed and unused.

"Traditional" processing of chromatographic data has relied on manual inspection of data to locate analytes of interest, followed by integration of peaks using an algorithmically determined baseline. While software to perform these analyses are readily available, this approach may require substantial user interaction, which can be time and resource intensive (Isaacman-VanWertz et al., 2017). Furthermore, the algorithms implemented in these software show limited capability in handling separation of co-eluted peaks, which leads to suboptimal utilization of data (Johnsen et al., 2013) and makes it difficult to extract clean mass spectra of analytes for accurate identification. These challenges often result in discarding potentially valuable information, particularly in large datasets that main contain hundreds or thousands of chromatograms that need to be processed. As fast chromatography has improved and field-deployable gas chromatography has advanced in fields like atmospheric chemistry (Lerner et al., 2017; Zhao et al., 2013; Apel et al., 2003; Goldan et al., 2004; Hornbrook et al., 2011), the size and complexity of chromatographic datasets make manual processing approaches unfeasible. Field instruments are also more impacted by shifts in operating conditions that may impact data reproducibility and peak co-elution due to non-ideal laboratory conditions (e.g., temperature fluctuations, etc.). Efforts to tackle the analytical challenge of integrating complex environmental datasets have focused on improved peak integration methods that use idealized mathematical peak shapes and defined mass spectra to resolve and integrate even poorly resolved chromatographic peaks (Blaško et al., 2009; Di Marco and Bombi, 2001; Isaacman-Vanwertz et al., 2017; Jeansonne and Foley, 1991; Mydlová-Memersheimerová et al., 2009; Naish and Hartwell, 1988). However, these methods still require manual inspection of the data to identify and catalog peaks of interest.

To facilitate peak identification in complex samples, matrix decomposition methods have been proposed resolve complex co-eluting peaks. In a relatively simple form, the co-variance of ions with chromatographic time can resolve representative mass spectra that rise and fall together as chromatographic peak. This approach has, for instance, been implemented as the

Automated Mass Spectral Deconvolution & Identification System (AMDIS) (Zhang et al., 2006) and is a useful tool for the

identification of analytes within a single chromatogram (Meyer et al., 2010). However, large chromatographic datasets present an opportunity to include an additional dimension of resolution, as not all chromatograms necessarily contain all the same analytes. As an example, a peak that is unresolved from a neighboring peak in one chromatogram may not exist in another sample, which would allow identification of the neighbor, and the neighbor could then be integrated more accurately in the first chromatogram now that its spectrum and retention time (and potentially peak shape) is defined.

Several multi-dimensional co-variance or factorization approaches have been developed to identify peaks across multiple chromatograms. One such approach is PARAFAC, a generalization of bilinear principal component analysis (PCA) to high order arrays (Hubert et al., 2012) in which a data array consisting of multiple chromatograms is decomposed into loadings and scores representing chromatographic profiles and mass spectra that can be more efficiently integrated. With relatively high signal-to-noise ratio and the proper number of components, a unique solution that consists of true mass spectra of analytes can

be found (Skov and Bro, 2008). PARAFAC2 was further developed to perform similar decomposition but with more robust handling of potential retention time shifts by not requiring all samples to have nearly identical time profiles, as PARAFAC requires (Zhang et al., 2014). Similarly, positive matrix factorization (PMF), a matrix decomposition method based on a weighted least squares fit (Paatero and Tapper, 1994), has been applied to deconvolve chromatographic data (Zhang et al., 2014). Unlike PCA, the result matrices, scores, and loadings of PMF are constrained to be non-negative, which reflects the

characteristics of environmental data more accurately (Paatero, 1997). Another major difference is that, in contrast to PCA, the factors obtained by PMF are not constrained to be orthogonal, are determined independently, and do not form a hierarchy in which each successive factor captures less variance. These advantages make PMF well-suited to describe environmental data, and PMF has become a preferred matrix size reduction technique, particularly in the field of atmospheric chemistry (Ulbrich et al., 2009). Prior applications to chromatographic data have instead focused on either the resolution and integration

of major components (Amigo et al., 2010; Hoggard and Synovec, 2007), or the extraction of average mass spectra and chromatographic profiles that provide binned information on broad classes of compounds (Zhang et al., 2014). However, this approach has some limitations, as minor components may provide unique information regarding the sources or chemical transformations of a sample.

This study presents an automated approach of cataloging and potentially identifying all analytes in a complex chromatographic

dataset. By coupling PMF, an established factor analysis technique with a decision tree, the approach described here deconvolutes complex chromatograms into a list of all or nearly all unique analytes and their associated mass spectra. This technique complements the new generation of tools described above developed to improve the efficiency and accuracy of integrating a list of known chromatographic peaks.

## 2 Methods

Our approach to cataloging all analytes in a set of chromatograms consists of two major processes. The first process is PMF analysis, and the second is a decision tree used to filtering, sorting, and cataloging PMF outputs into a list of analytes. We first provide an overview of the PMF algorithm before describing the overall cataloging approach. In this work, the term "analyte" is used to refer to a chromatographic peak (e.g., a chromatographic "feature") with unique mass spectrum and retention time whether it has a known definitive identification or not, following the usage of this term other studies (Amigo et al., 2010;
Grace et al., 2019; Isaacman-VanWertz et al., 2017; Li et al., 2022).

### 2.1 Positive matrix factorization

Positive matrix factorization is a bilinear model that approximates an observed data matrix, X, by finding the weighted least squares solutions for a set of factors that can describe the dataset (Paatero and Tapper, 1994). In the case of chromatographic data, the data is a matrix in which rows represent the average mass spectra of each averaging time period (typically 1-5 Hz)
and columns are the time series of each mass spectral mass-to-charge ratio (*m/z*). The model is represented as:

$$X = GF + E \tag{1}$$

where X is the data matrix, E is the residual matrix, and G and F are the score and loading matrices, respectively. The chromatographic data matrix is thus described by a set of factors, each of which has an average mass spectrum (the loading) and a time-dependent score that represents the chromatographic profile. The elements of both G and F matrices are constrained
to be non-negative and are therefore expected to more accurately represent real-world data than PCA or other PCA-based matrix decomposition methods.

The number of factors is prescribed *a priori* by the user, which represents a major source of uncertainty and subjective interpretation in typical PMF applications. In contrast to other PMF applications, the primary goal in this work is not to optimally describe the complete data set, but rather to increase the number of factors to a point where even minor components
are extracted as separate factors, even at the risk of over-fitting the data (which will be rectified by a subsequent decision tree). With this approach, any existing analytes with a significant level of signal should be identified as separate analytes, regardless of whether such an analyte is a compound present in the sample or is a contaminant. Assigning too many factors (e.g., greater than the number of compounds in the sample) in the model can cause "factor splitting," a phenomenon in which a factor that might carry real-world meaning or interpretation is divide into multiple factors that cannot be readily interpreted (Hoggard and
Synovec, 2007). In the context of this work, splitting would result in the separation of an analyte into multiple different chromatographic peaks that all represent the same analyte. We address this case using the decision tree presented below and therefore do not rely on existing metrics for evaluating the optimality of the factor solution (e.g., the ratio of error in the solution to expected error, "$Q/Q_{exp}$").

In this study the PMF Evaluation Tool, PET version 3.04 (Ulbrich et al., 2009) software package in Igor Pro 8 (WaveMetrics,
Inc.) was used to run the PMF2 (Paatero and Hopke, 2009) algorithm on the dataset and obtain PMF outputs. Chromatograms

were analyzed using the freely available TERN software package within the same programming environment (Isaacman-VanWertz et al., 2017).

## 2.2 Description of analyte cataloging method

### 2.2.1 Batch process of positive matrix factorization

An overview of the complete process is shown in Fig. 1. Multiple chromatograms representing time-varying mass spectra are stacked to yield a three-dimensional data array (I x J x K), where I and J constitute a chromatogram (elution profile x mass spectra) and K is the number of samples (Amigo et al., 2010). Each chromatogram is first aligned to the same retention time basis by using a small number of known compounds or introduced standards in each sample to define known retention times. Strictly speaking, this pre-processing is not necessary for factor analysis. However, interpretation of the outcome of data

reduction techniques such as PARAFAC(2) and PMF can be unreliable when chromatograms are used directly as input (Eilers, 2004; Van Nederkassel et al., 2006), as it may be difficult or impossible to determine if unaligned peaks in each chromatogram represent the same analyte. Chromatogram alignment may occur through manual adjustment by users or may be automated using any of multiple solutions (Eilers, 2004; Kassidas et al., 1998; Nielsen et al., 1998) to align chromatograms in the pre-processing of data with relatively little user input. Implementation of some alignment (manual or automated) is necessary in

pre-processing, but the cataloging approach described here is independent of the details of any such approach (a manual approach is used in this work), so details are not included. To achieve high chemical resolution and identify minor constituents, PMF is not performed on the full matrix, but rather on sub-sectioned "slices" that represent short periods of the chromatogram. As samples of environmental data can be highly complex and heterogenous, applying PMF to small portions of the elution time successively is significantly more effective than extracting factors from the full range of the elution time data (Zhang et

al., 2014). Each slice is comprised of the same period of elution time from each sample chromatogram. Though retention times of chromatograms have been globally aligned, some small shifts in retention time may still exist at the timescale of a slice; consequently, a secondary, "fine-scale" retention time correction is applied to each sample within the slice to yield a unified time basis (typically the retention time of one of the chromatograms). A form of correlation optimized warping (COW) (Nielsen et al., 1998) is used, in which the retention time offset is determined that maximizes the correlation of the maximum number

of single ion chromatograms (SIC) within the slice. This approach is expected to work so long as a dominant fraction of peaks are present in all chromatograms but does not require all peaks to be present. This fine-scale retention time adjustment is not strictly necessary for the application of PMF; without it, the same analytes will generally be found in the chromatograms. However, minimizing retention time differences between chromatograms is very useful for the subsequent decision tree to determine whether peaks in different chromatograms represent the same analyte, as opposed different analytes with highly

similar retention times and mass spectra (e.g., isomers).

For each slice of the three-dimensional data array, the dimensionality is reduced by concatenating samples, such that the first row of the two-dimensional matrix of one chromatogram is positioned after the last row of the matrix of another chromatogram.

The resulting two-dimensional slice matrix represents repeating periods of elution time. PMF is then applied to this concatenated slice matrix, yielding a set of factors that represent the elution profiles of mass spectra that co-vary (i.e.,

chromatographic peaks of analytes). For the reasons discussed above, tens of factors are used in the PMF solution, which is higher than used in most other more common PMF applications (Ulbrich et al., 2009). This number of factors is on the order of the length of the slice divided by the typical peak width (i.e., one or two factors per resolvable peak), but a more detailed discussion of optimizing the number of factors is discussed in the results section. Each resulting PMF factor from the slice consists of the chromatographic profiles of a given mass spectrum in each chromatogram (an example is shown in Fig. S2).

Factor data are stored for subsequent processing, and PMF is performed iteratively through slices until the entire range of chromatographic time is covered. Each slice overlaps in part with the slices before and after to capture potential peaks that may be cut off at the edges of each slice; overlap must equal or exceed the typical width of a chromatographic peak to ensure this outcome. The full set of PMF results for all slices are compiled and addressed through a decision tree, shown broadly in Fig. 1 and addressed in more detail below.

The optimal number/length of slices is expected to be data-dependent and is expected to control the number of factors needed to fully deconvolute the data. Shorter slices can likely be analyzed with fewer factors, but require more slices to analyze the full chromatogram, making estimation of computational time somewhat complex. In this work we use slices of 5-15 seconds and PMF solutions with up to 35 factors, then discuss tradeoffs in the results. We demonstrate in this work that the decision tree described below addresses potential factor splitting caused by high factor solutions, so there is little disadvantage to

increasing the number of factors other than the additional required computational time. In contrast, decreasing the number of factors can result in the detection of fewer analytes. Consequently, it is necessarily a decision of the user of this method to balance potential decrease in data extraction against computational resources.

### 2.2.2 Decision tree

A detailed description of steps involved in the decision tree is presented in Fig. 2. Steps are described in detail below, but the

overall approach proceeds as follows:

(1) Peaks are detected in each PMF factor and cataloged by quantitative parameters describing their idealized mathematical form.

(2) Spurious peaks are removed by several filters to eliminate noise.

(3) Peaks in each factor are sorted into potential analytes based on retention times.

(4) Potential analytes are sorted and combined into a catalog of unique analytes by comparing retention times and mass spectra.

*Peak detection and fitting*. Within a factor, peaks are detected by using first and second derivatives to find local minima and maxima. All found peaks are then simultaneously fit to mathematically idealized forms; we assume a Gaussian curve as the ideal chromatographic peak shape (Anderson et al., 1970), though experimental peaks are often perturbed from the ideal shape

by instrumental factors and a certain level of mixing of signals is introduced (Isaacman-VanWertz et al., 2017). Implementation

of this approach was performed using built-in packages within the Igor Pro 8 programming environment (specifically, "Multipeak fitting 2"). A more complex approach could include modified peak shapes (e.g., convolution with an exponential (Isaacman-VanWertz et al., 2017)), which would likely enable more accurate characterization of the parameters describing a peak. However, in this work the goal is to catalog all peaks by their approximate parameters as opposed to perfectly integrate them, so increasing the complexity of peak fitting by incorporating refined peak shapes has not been implemented. Implementation of exponentially modified gaussian (EMG) as a peak fitting model has been inspected on the samples containing deuterated tetradecane presented in Fig.4 and discussed in the supplementary information (Fig. S7). Optimal peak shapes could be used in subsequent processing for accurate integration of data. The outcome of this peak fitting is a set of peaks present within each factor chromatographic profile (example shown in Fig. S3), including their known retention times within each chromatogram and their retention times relative to peaks in all the other chromatograms (i.e., retention times both uncorrected and corrected to a unified time basis). Not all factor chromatographic profiles necessarily contain any peaks at all, with noise or background factors frequently returned (as seen in the example shown in Fig. S2). The parameters that quantitatively describe the peak (location, width, and height) are stored alongside corresponding time profiles and mass spectra from the PMF. The location of a peak (i.e., retention time) is the mean of the Gaussian curve, and the width of a peak is described using the standard deviation of the curve. The uncertainties in these three parameters determined by the fit are also stored. Alternate descriptors of width such as full (or half) width at half maximum, FWHM (or HWHM) may be more appropriate if other peak shapes are considered, but all of these descriptors can be mathematically related for a Gaussian curve, so any descriptor is useful in this case. We discuss the implications of this choose in the discussion of analyte sorting. By performing peak detection and initial fitting on all factors within the slice, an index of all potential peaks is generated within the region of the chromatogram. These peaks are then validated and cataloged into analytes in subsequent steps by a decision tree.

*Peak filtering*. Once all potential peaks are indexed, spurious potential peaks are eliminated. The automated peak detection and fitting algorithm may find peaks in factors that do not qualitatively appear to contain any chromatographic peaks or may imply that the inclusion of a negative peak improves the fit (Anderson et al., 1970). Filtering thresholds are introduced to remove these peaks. Specifically, peaks with negative parameters and/or an estimated error greater than the corresponding parameter (i.e., width) are considered to be a result of a bad fit and thereby rejected. In addition, peaks with weak signal (peak height/baseline signal < 10) are removed. Furthermore, because true chromatographic peaks are expected to have a peak widths that can be reasonably well-defined (i.e., the range of possible widths in a dataset may be variable but is generally not very broad), outliers of peak widths are identified using Tukey's fences method (Tukey, 1977) with a conservative range for reasonable peak width. The range is defined as:

$$[Q_1 - k(Q_3 - Q_1), Q_3 + k(Q_3 - Q_1)] \tag{2}$$

where Q1 and Q3 are lower and upper quartiles of the sorted peak width, respectively, and k = 3. Peaks with widths outside this range are rejected, i.e., those with widths outside the interquartile range by more than three times the magnitude of the interquartile range. Finally, a small number of peaks whose parameters are near both the upper boundary of the peak width

and the lower boundary of the peak height to base ratio (i.e., low abundance, broad peaks, those with a height/width ratio of, empirically, ~10000) are rejected as they indicate either poor fitting of the peaks or a fitting of noise.

*Peak sorting*. Ideally, using the optimal number of factors will result in each factor representing one chemical compound, but in practice more than one analyte may be detected within a given factor. This may be due to the true presence of multiple analytes within the slice that have mass spectra too similar to deconvolve (e.g., branched alkane isomers), or may be due to an

error introduced during peak detection or fitting. Peaks in a given factor by definition share a mass spectrum, so those that are chromatographically separated by less than a critical retention time difference (i.e., have nearly the same retention time) are assumed to represent the same analyte. Selection of a critical retention time difference is somewhat dependent on the goals of the user but is inherently related to peak widths. A conservative estimate of a critical width is several times the standard deviation (e.g., FWHM = 2.355σ), which would ensure that only peaks that are truly chromatographically resolved are regarded

as unique. However, in many cases, isomers may not be well resolved but nevertheless represent unique analytes, which may be apparent in small changes in ion ratios or signal intensities across chromatograms. In these cases, a more aggressive (i.e., smaller) approach to critical retention time differences may be appropriate, which might include HWHM (~1.18σ), or, most aggressively, peaks that are separated by only one or two datapoints (i.e., a peak in a different time period of instrument acquisition). Setting this parameter more aggressively increases the possibility of positive errors, discussed in Section 2.4.

When peaks with the same mass spectrum (i.e., from the same factor) closer together than critical peak width are found in the same chromatogram, they are assumed to be the product of factor and/or peak splitting and are combined. Peaks that meet these criteria across multiple chromatograms (i.e., found within one factor at the same relative retention time) are assumed to represent the same unique analyte across each chromatogram. Peaks within a factor that are separated by more than a peak width are considered unique analytes. This process sorts the peak catalog to yield a list of all potentially unique analytes across

all factors, with some factors containing multiple analytes (all with the same mass spectrum) and some factors containing no analytes. Potential analytes each have an associated retention time and mass spectrum, and a known peak height and width in each chromatogram in which it was found.

It is theoretically also possible that multiple analytes are present in a factor not because they have similar spectra, but because the number of factors is substantially lower than the number of analytes present so PMF yields some approximate convolution

of analytes. However, in practice, this issue is largely avoided by using a large number of factors. Furthermore, the approach to combining peaks with similar retention times and spectra within a chromatogram may capture a small number of isomers that share a mass spectrum, are rarely resolved, and covary between samples, (e.g., *m*- and *p*-xylene). This limitation is likely inherent to the approach, as their resolution by an automated peak detection algorithm would require an assumption of peak shape, which would limit its application to complex data. Isomers such as these represent an example of the potential impact

of a user-specified critical retention time difference, as an aggressive value (e.g., one or two datapoints) may separate these analytes if there is at least some separation by retention time and some variability in ratios between samples that may be detected by the PMF, while a more conservative approach (e.g., FWHM) is unlikely to separate poorly resolved isomers.

*Analyte sorting*. PMF followed by peak fitting, peak detection, and peak sorting is performed for all slices, generating a list of potential analytes across the full chromatographic range. Because these potential analytes were generated by examining peaks within each factor, this process does not account for the possibility that PMF factor splitting generates multiple factors containing the same analyte with slight variations in their mass spectra due to instrument drifts, etc. To address the issue of factor splitting, all the potential analytes are intercompared to remove and combine possible repeats. Mass spectra (i.e., the mass spectrum of the factor in which they were found) are compared by cosine similarity:

$$\epsilon = \frac{M_1 \cdot M_2}{||M_1|| ||M_2||} \tag{3}$$

where $M_1$ and $M_2$ are normalized mass spectra of two potential analytes being compared to determine whether they represent the same analyte. This is the preferred approach of commonly used mass spectral libraries and search programs (Stein, 2014). Two identical mass spectra will have $\epsilon = 1$. Values of 0.8 and higher are generally considered to indicate two mass spectra that may represent the same analyte (Stein, 1994; Worton et al., 2017).

Analytes with mass spectral cosine similarity values, $\epsilon \geq 0.8$, are compared by their retention times. In cases where the difference is greater than the median width – in other words, if the peaks are considered sufficiently distant to each other – they are cataloged as two unique analytes. Analytes with matching mass spectra and retention time differences below the critical threshold are considered to be the same compound detected by two different factors or are two analytes that cannot be resolved by the instrument either chromatographically or by their mass spectra. When found within one slice, analytes are combined by summing peak heights and weighted averaging of their defining parameters (width, mass spectra, etc.). In overlapping sections between slices, any repeat analytes (i.e., found in both slices with matching retention times and spectra) are simply filtered out. Again, the selection of the critical retention time difference exerts some control on the opposing tendencies of this approach to either consider peaks unique (potentially leaving multiple peaks representing the same analyte) or combine peaks (potentially binning multiple analytes). In this step, any potential analytes being compared must exhibit at least some difference in mass spectrum and sample variability, since they were separated by the PMF, so a more aggressive critical retention time difference is likely warranted here.

The outcome of this analyte sorting process is a catalog of unique analytes with associated retention times and mass spectra, including information about their widths and heights in each chromatogram used in the analysis. Examples of analytes found are provided in Figs. S4 and S5, which are discussed in more detail in the results section below. This catalog of analytes is the end goal of the present work, but could be used as a template for subsequent analyses, or as a dataset to be matched against existing libraries or authentic standards for identification (Worton et al., 2017).

### 2.3 Sample datasets

The method developed is tested using two GC-MS datasets: a laboratory-generated dataset of known standards and a dataset of ambient aerosol samples. Both datasets were collected using a Semi-Volatile Thermal desorption Aerosol Gas chromatograph (SV-TAG). This instrument has been described elsewhere in detail (Isaacman et al., 2014; Williams et al.,

2006; Zhao et al., 2013). In brief, sample is collected on a passivated metal fiber filter housed in a temperature-controlled cell, either by introducing a liquid standard or pulling through sampling ambient air. The cell is then thermally desorbed with a programmed temperature ramp (25 to 315°C over 8 minutes) and analytes are transferred to a GC column ramped from 50°C to 310°C. GC eluent is then analyzed by electron ionization mass spectrometry (Agilent Technologies). The two datasets differ in their column ramp rate and dimensions. Laboratory data was collected with a MTX-5 column (15m × 0.25mm × 0.25μm, Restek) at a ramp rate of 12.5°C/min. Ambient data was collected with a Rtx-5Sil MS column (20m × 0.18mm × 0.18μm, Restek) at a ramp rate of 23.6°C/min.

For analysis of known standards, liquid standards were injected into the sample collection cell through the automated liquid injection system of the TAG (Isaacman et al., 2011). Standards included 10 ng of $n$-alkanes ($C_8$-$C_{40}$ diluted from 500μg/ml, supplied by AccuStandard) and 15ng of select perdeuterated n-alkanes: $C_{14}$, $C_{15}$, $C_{16}$, $C_{20}$, $C_{24}$, and $C_{26}$ (diluted from stock mixtures made of pure compounds, supplied by C/D/N Isotopes approximately two years prior to use).

Collection of ambient air data took place near Manaus, Brazil as part of the GoAmazon2014/5 campaign (Martin et al., 2016). Details of sampling and the SV-TAG instrument used to collect this dataset has been previously published (Isaacman-VanWertz et al., 2016). The data presented in this work were collected during the wet season in February and March 2014. Samples of atmospheric particles and semi-volatile gases were collected during the first 22 minutes of every hour. During desorption of the collection cell, analytes were derivatized by introducing N-methyl-N-trimethylsilyltrifluoroacetamide (MSTFA) into the desorption flow; this method silylates all hydroxyl groups, improving transfer through the GC column (Isaacman et al., 2014). Approximately 100 compounds have been previously resolved and cataloged in this dataset (Isaacman-VanWertz et al., 2016), of which only a small fraction were identified as compounds with known molecular structures and identities.

**2.4 Method validation**

The analyte cataloging method is investigated using real-world GC-MS data collected on known calibrants and under field conditions, as described below. Two major failure modes are examined: (1) negative errors in the form of uncatalogued analytes due to underfitting, and (2) positive errors in the form of false analytes identified due to factor splitting or overfitting. The former can theoretically be addressed in large part by increasing the number of factors, but this approach increases the potential for the latter. To examine this interplay, and the ability of the decision tree to compensate for potential positive errors, we examine sections of chromatograms containing known $n$-alkanes and perdeuterated isotopologues. These samples are analyzed with a varying number of factors to understand the ability of the method to identify major and minor components, address factor splitting caused by high numbers of factors, and examine the potential impacts of the critical retention time difference. Slices of 4 chromatograms of a 5-15 second window containing known analytes are investigated under a range of method parameters. Application to complex field data provides an addition test for negative errors by challenging the method with data that has previously been catalogued by an expert operator, as well as providing insight into the power of the proposed method.

To validate the method, we discuss below the results of three specific tests. In the first test (Section 3.1), we investigate the potential for positive errors by using high-factor PMF solutions to generate the catalog of peaks used by the decision tree. In the second test (Section 3.2), we investigate the potential for negative errors by examining the deconvolution of poorly resolved analytes with similar mass spectra. In the third test (Section 3.3), we investigate the utility of the method in real-world data by applying the method to a complex environmental sample and examine the potential for negative errors by comparing the analyte catalog to a previously published analysis (Isaacman-VanWertz et al., 2016).

## 3 Results and discussion

### 3.1 Effects of increasing factors

The proposed cataloging method was applied to a 15-second chromatographic window that included the peak known to represent injected perdeuterated tetradecane ($C_{14}D_{30}$), with the number of PMF factors ranging from 1 to 20 (Fig. 3). In a one-factor solution, one analyte was found, representing the known compound (Fig. 3a). The number of analytes found increased as more factors were used, with the injected compound always found and minor analytes found in higher factor solutions as discussed below. The critical retention time difference used in this analysis was relatively aggressive (median HWHM, which equals 0.7s in this data. in order to examine the capability of the method to find unique peaks; the effects of this selection are discussion below. The relationship between the number of analytes found and the number of factors is non-linear, approaching an apparent plateau. This plateauing behavior for analytes is in contrast to growth in the number of peaks found, which continues to increase linearly with number of factors. An ever-growing number of analytes is physically improbable given the relative simplicity of the data, and these peaks likely represent factor splitting. The decision tree addresses this issue by rejecting and combining these found peaks, eventually yielding six unique and distinct analytes that remain relatively stable across solutions.

This result agrees with the trend in the percent of the total signal that is not described by the found analytes, i.e., percent residual calculated as the sum of absolute difference between the total ion chromatogram and the reconstructed signal curve at each point in time relative to the sum of total ion signal. With increasing factors, percent residual first drops, from approximately 17% with one factor down to less than 10% with a few factors. Though identifying the main injected compound and describing 83% of the measured data is independently quite compelling, the 9-factor solution (Fig. 3b) suggests that the measured data can be better described by increasing the number of factors. Though these analytes appear to represent splitting of the chromatographic peak, we demonstrate in the following section that these data represent real analytes that might be overlooked by a manual operator. The 3 to 4 major analytes are therefore cataloged with only a few factors, and the same major analytes are cataloged in a 9-factor solution as a 19-factor solution (Fig. 3c). Subsequent increases in the number of factors, from 4 to 20, yield little additional information to describe the measured data, detecting only a small number of low-abundance peaks. Overall, these results demonstrate that the approach can identify minor components, while increasing the

number of factors beyond the minimum necessary neither provides additional information nor impedes the method (other than the additional computational resources used).

## 3.2 Deconvolution of poorly resolved analytes

Analysis of perdeuterated tetradecane discussed above indicates that most of the signal can be described by 3 analytes and while a few additional analytes may be present, their inclusion does little to describe the overall signal. These 3 analytes are those shown in Figs. 3b and 3c as poorly resolved peaks, and they are commonly identified in all factor solutions that found three or more analytes (i.e., 4-factor solutions and higher). The presence of 3 analytes under this peak is curious, as the known sample was comprised of only perdeuterated tetradecane ($C_{14}D_{30}$). However, we demonstrate here that these data can be described as two isotopologues, $C_{14}D_{29}H$ and $C_{14}D_{28}H_2$. (Fig. 4), which are not unexpected in isotopically labelled standards mixed into methanol, in particular those that were purchased several years prior as is the case here.

The total ion chromatographic peak appears to be normally distributed with minimal skew, but the analyte cataloging method we develop here finds three analytes with slight shifts in retention time and differences in mass spectra. One analyte is deconvolved using only a 2-factor solution (Fig. 4a) and the third is found in higher-factor solutions (Fig. 4c). Retention times are shifted later, as expected for replacement of a deuterium with a hydrogen, and projection of this trend forward (i.e., replacement of all deuterium with hydrogen) predicts a retention time roughly that of non-labelled tetradecane as expected. Similarly, the fragmentation patterns are highly similar, but there are some significant differences in their intensities. This is clearest in the fragmentation patterns at their molecular weight, with $C_{14}D_{30}$, $C_{14}D_{29}H$, and $C_{14}D_{28}H_2$ having substantial signal at $m/z$ 228, 227, and 226, respectively. At lower $m/z$, all compounds have large signal on mass $m/z$ 66 and differences of 16 ($CD_2$), but isotopologues also have higher signal at masses shifted by one or two amu (e.g., higher m/z 65 for $C_{14}D_{29}H$). The separation of isotopologues presents one of the most difficult challenges for separation methods like chromatography (Valleix et al., 2006; Filer, 1999) due to the tendency of isotopologues with relatively smaller signal to be completely embedded in the peak of their counterpart and their similar spectral signals (Amigo et al., 2010). Figure 4 demonstrates these issues, and the ability of the method to overcome them.

It is a clear possibility that the deconvolution of these three analytes is a case of positive error – that these found analytes are an error within the method as opposed to real analytes. To test for that possibility, we performed the same analysis on non-labelled tetradecane and found all signal reasonably described by a single analyte with no co-eluting analytes as expected (Fig. S6). This result supports the conclusion that the additional peaks found for deuterated alkanes are not artifacts due to the high factor solutions but rather represent true co-eluting peaks that demonstrate the ability to find difficult-to-resolve analytes.

Separation of these isotopologues presents an opportunity to examine the impact of the critical retention time difference, and the impact of assumed Gaussian peak shapes on this separation. Though exhibiting interpretable differences in their higher molecular weight ions, the heavy fragmentation of alkanes yields mass spectra that are not sufficiently different to be separated by the cosine similarity threshold (i.e., comparisons between all three isotopologues have $\epsilon \geq 0.8$), despite sufficient differences to be separated into different factors in the PMF. Consequently, resolution of these peaks relies on separation by

retention time in the analyte sorting step. Separation between each peak is roughly 0.75 second in retention time, while median peak width in the dataset ($\sigma$) is 0.6s, peak widths of these analytes are roughly on the order of 0.7s, and a mass spectrum is collected every 0.3s. Peaks are consequently separated by more than two datapoints, and more than the median HWHM of the dataset (0.71s), but not by more than the HWHM of these specific peaks (0.82s) or by more than the median FWHM of the dataset (1.4s). In other words, only more aggressive screening methods (i.e., using $\sigma$ or median HWHM as the critical retention time difference) would separate these isotopologues. This approach also increases the chance of chromatographic artifacts being cataloged as real analytes (positive error), but a more conservative approach increases the possibility of overlooking poorly resolved and similar analytes such as these (negative error). Ultimately, it is up to the user to decide the optimal critical retention time difference.

The effect of a non-Gaussian peak shape was also examined. Because peak detection relies on derivatives to identify potential peaks based on inflection points in the data, the number of peaks found is agnostic toward peak shape; instead, peak shape primarily impacts peak widths. Using an exponentially modified Gaussian peak shape to the analysis of isotopologues does not substantially change the result (Fig. S7). With this peak shape, isotopologues remain separated using more aggressive critical retention time differences (median HWHM or more than two datapoints) but are combined by more conservative thresholds. This result is of course limited to the shown case, in which a Gaussian curve reasonably describes the observed data. Datasets containing highly non-Gaussian peak shapes may be more impacted and should be examined closely for the potential impact of peak tailing on positive errors.

### 3.3 Cataloging analytes in real-world data

To evaluate the proposed method in a real-world application, we apply it across the full chromatographic range for data representing the gas- and particle-phase composition of atmospheric samples. The goal of this analysis is to both provide an estimate of the number of analytes found in representative atmospheric samples and evaluate the ability of the cataloging approach to identify analytes known to exist in a complex, real-world dataset. Doing so requires user decisions on the optimal parameters (e.g., number of factors, slice length). Figure 3 demonstrates the tendency of the method to find increasing numbers of analytes with increasing factors until reaching a certain threshold. It is reasonable to expect that the maximum number of analytes found in each slice is also a function of slice size (i.e., length of the chromatographic window). As slice length increases, the number of slices decreases (roughly linearly scaling computational time) and the number of necessary factors increases (roughly exponentially scaling computational time, Fig. S9). Because the decision tree is effective at mitigating positive errors, the results of the methods (i.e., the catalog of analytes) are not strongly impacted by optimization decisions, which instead primarily impact efficiency (i.e., minimizing computational time). For the real-world data tested here, the maximum number of analytes observed in each slice roughly approaches a plateau when the number of factors used is 2-3 times the length of each slice (in seconds) (Fig. S8). Due to the balance between factor number and length, it is generally somewhat more efficient to use lower-factor solutions for a larger number of shorter slices, but computational time is not

substantially different across different sets of parameters that meet the necessary number of factors per slice length (Fig. S10). For this data, we use a 25-factor PMF on 10-second slices with 2-second overlap between slices (based on a typical peak width of 1-2 seconds).

A sample of four chromatograms within the retention time window 200 – 650 second were analyzed by this approach (Fig. 5), constituting 56 slices. These chromatograms were selected from the complete dataset based on their similar sampling times, minimizing differences in instrument operating conditions (retention time, mass spectrometer tuning) over time. We recognize that the variance obtained by a larger sample size may increase the amount of information extracted, however this would significantly increase the computational expense. In essence, the optimization of sample size is dependent on sample-to-sample variability and processing capability. In this work, we use a sample of four chromatograms to demonstrate the effectiveness of this approach, and optimization of sample size is dataset-dependent and will be explored in future work. From this sample of chromatograms, a total of 1169 analytes were identified, with a computational time of 290 minutes. This analysis uses a moderately aggressive critical retention time difference (1.4$\sigma$), but the number of analytes found is slightly reduced by more conservative approaches (e.g., only 20% lower at using the much more conservative FWHM, Table S1). In contrast, a previously published analysis of this dataset focused on only ~100 compounds cataloged by manual inspection though additional compounds are observed to exist in the dataset that were not a focus on this previous analysis. We note that a major advantage of the proposed approach is not only the larger number of analytes cataloged (with significantly less manual interaction), but also that each of these analytes has a well-defined mass spectrum that can be used for identification or comparison to existing mass spectral libraries. We probe the present analysis for negative errors by comparing the analyte catalog against analytes identified in the previously published analysis. The analyte catalog in this work was found to detect every peak in the dataset with a known identification, including introduced isotopically labeled internal standards, analytes identified by authentic standards, and tracer compounds of interest for known atmospheric processes such as oxidation products of naturally emitted gases and emissions from biomass burning. For example, the identified peaks observed in the inset of Fig. 5 at 350 and 356 seconds are the known, highly studied oxidation products of isoprene, 2-methylthreitol and 2-methylerythritol (Claeys et al., 2004; Surratt et al., 2010; Wang et al., 2005), while the peak co-eluting earlier at 350 seconds is the α-pinene oxidation product pinic acid. The mass spectra were also compared to the NIST library and only 96 (~8%) of the cataloged analytes had mass spectral matches in the library that were in the "good" or "excellent" range (Stein, 2008) (Fig. S11). Previous work has shown that matches below these thresholds indicate the found spectra does not represent the unknown analyte (Worton et al., 2017), suggesting that 90% of analytes in these samples do not exist in the mass spectral library. These results demonstrate both the utility of the proposed approach to detect and identify known analytes of interest, and to catalog hundreds of unknown analytes by their retention time and mass spectra. Significant work remains to be done to identify the unknown compounds in the atmosphere. However, many tracers commonly used by the community started out as components with unknown structure or origin. For example, C5 alkene triols that are commonly measured as isoprene oxidation tracers required significant dedicated effort to identify (Wang et al., 2005). Previous work has also been done wherein correlation to known tracers was used to identify the likely sources of unknown compounds (Isaacman-VanWertz et al., 2016), and in some cases,

this information was used to quantitatively attribute sources of aerosol (Zhang et al., 2018). Therefore, despite the lack of
current identification, we believe it is useful to integrate and investigate all analytes, and examine the data as a whole.

## Conclusions

In this work, we describe and evaluate a method to catalog analytes in a set of chromatograms representing complex environmental data. Analysis of known standards demonstrates high skill at finding minor analytes even when poorly resolved, with no strong tendency to find spurious analytes that are not actually present. This approach will consequently be valuable
for the automated processing of complex chromatographic data and will enable new information to be extracted that might otherwise be ignored or discarded by conventional approaches due to technical difficulties or limited resources. Analysis of real-world data cataloged more than 1000 analytes with little or no human interaction. Three major future developments would further enhance this approach: improved retention time correction without human interaction (e.g., by parametric optimized warping (Eilers, 2004)), incorporation of modified peak shapes (e.g., exponentially modified gaussian) as a peak fitting model,
and algorithmic optimization of decisions around the length of each slice, the number of factors, and the number of chromatograms. However, even without these advancements, the required levels of operator interaction are limited, and the proposed method has the potential to substantially improve and expand data analyses of both new and previously collected data.

## Author contributions

GIVW conceptualized and supervised the project; SK developed the software code, performed the analysis, and wrote the manuscript draft; BML, DTS, and GIVW contributed to the analysis, and reviewed and edited the manuscript.

## Competing interests

BML and DTS are employed by Aerodyne Research, Inc., which commercializes GC and MS instrumentation.

## Acknowledgement

This work was supported by the National Oceanic and Atmospheric Administration Small Business Innovative Research Program (WC133R18CN0064 and NA21OAR0210294). We thank Chenyang Bi for assistance with generating laboratory data and Allen Goldstein for sharing ambient data.

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

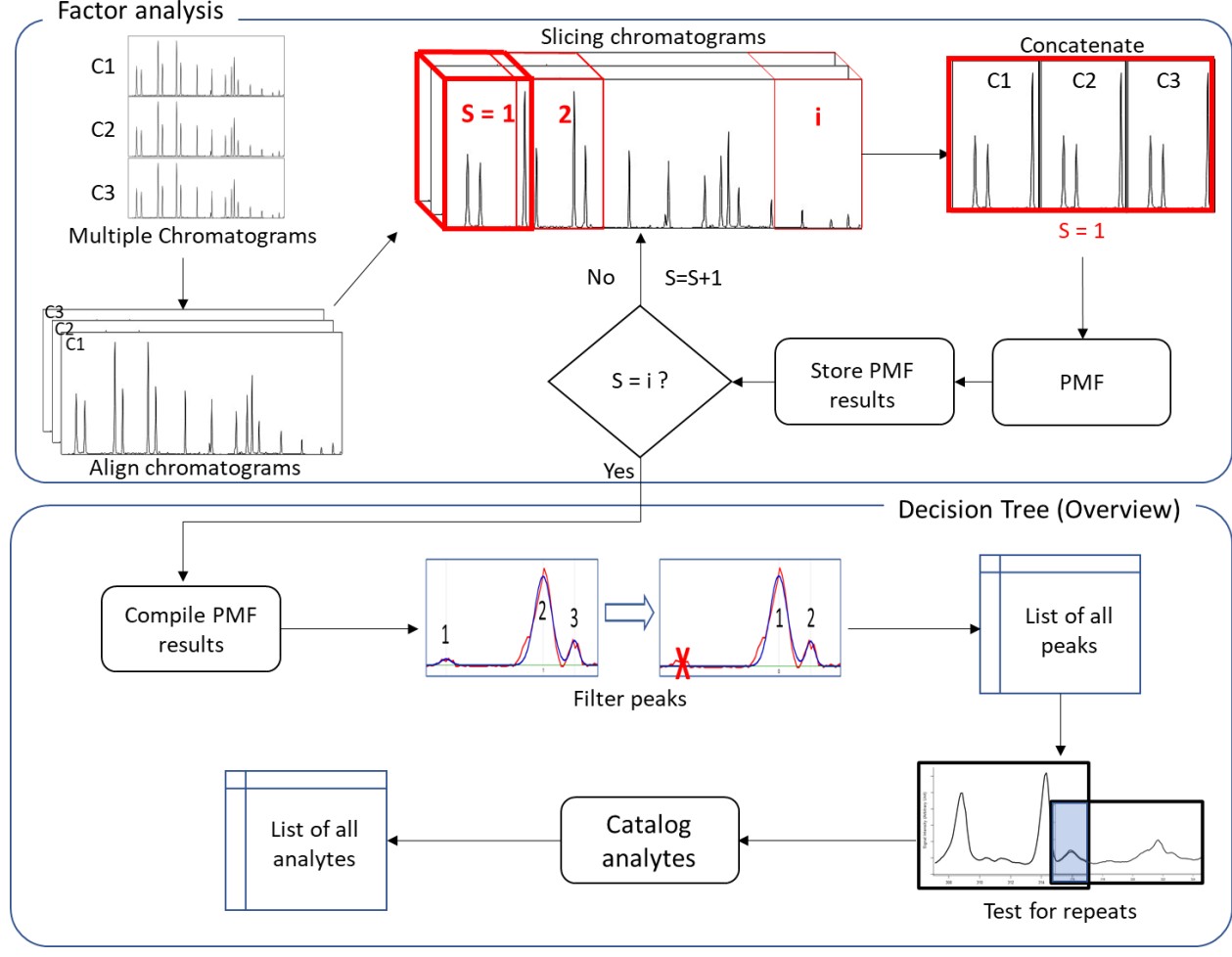

**Figure 1: Flowchart of analysis process including an overview of the developed decision tree implemented in this study. Chromatograms and subsections are labeled as C and S, respectively.**

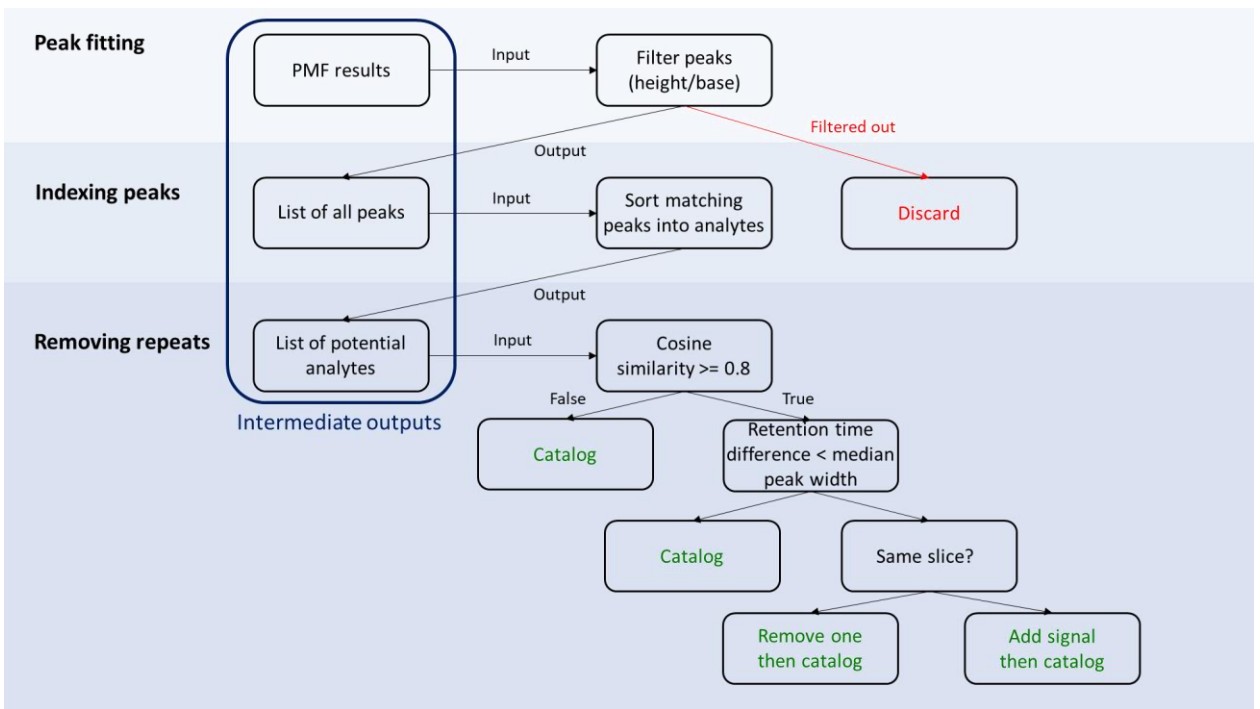

**Figure 2: Detailed work flow of the decision tree. Steps involved in cataloging individual analytes are displayed in green.**

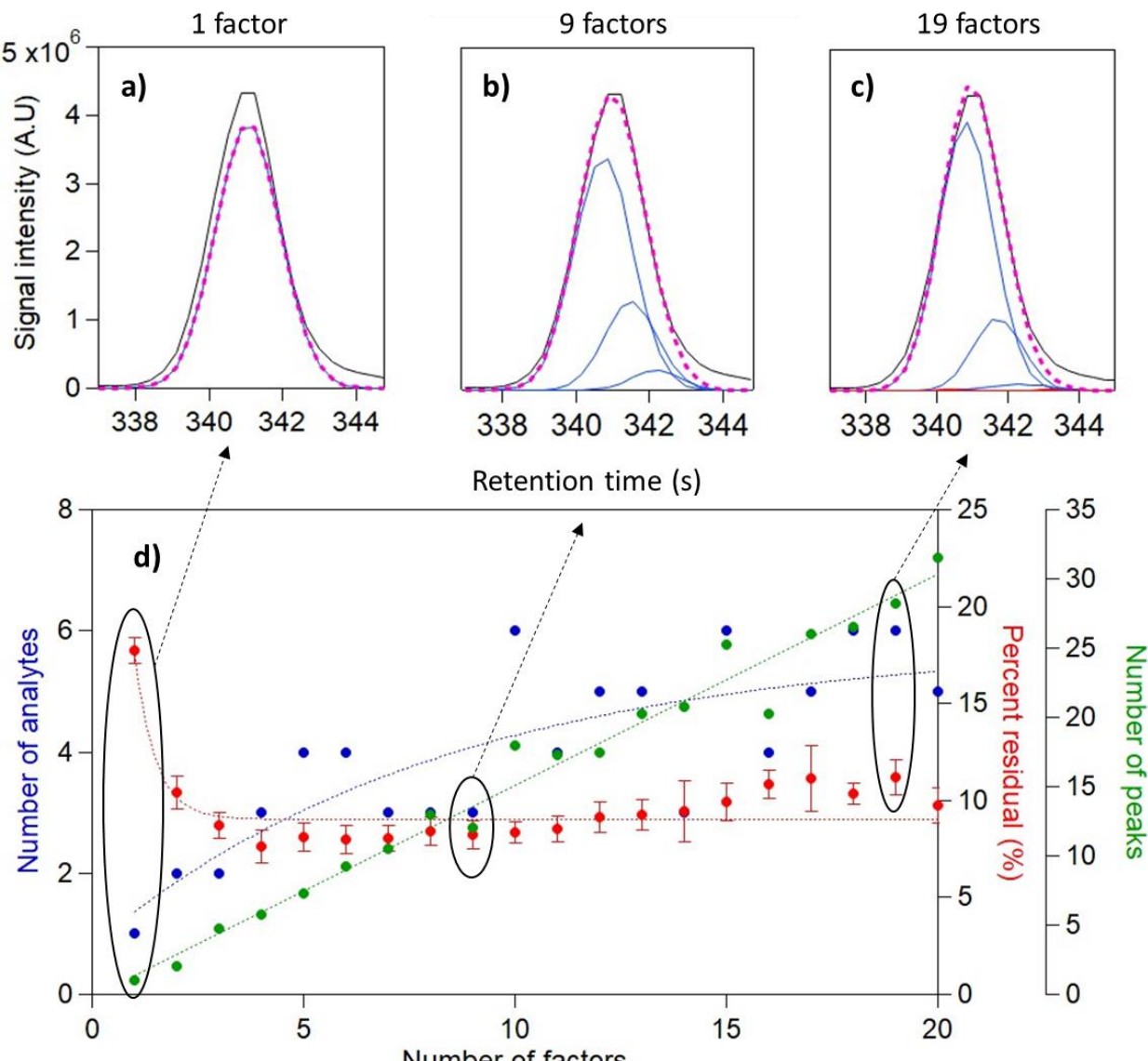

**Figure 3: Analysis results of n-alkane mixture with four chromatograms and a 15-second time window containing perdeuterated tetradecane over a range of factors (1-20). Total ion signal and the reconstructed total signal of (a) 1-factor, (b) 9-factor, and (c) 19-factor solutions are displayed above. The number of analytes identified and calculated percent residual over varying number of factors are displayed at the bottom (d) in blue and red, respectively. Error bars represent the standard error of percent residual values from all chromatograms.**

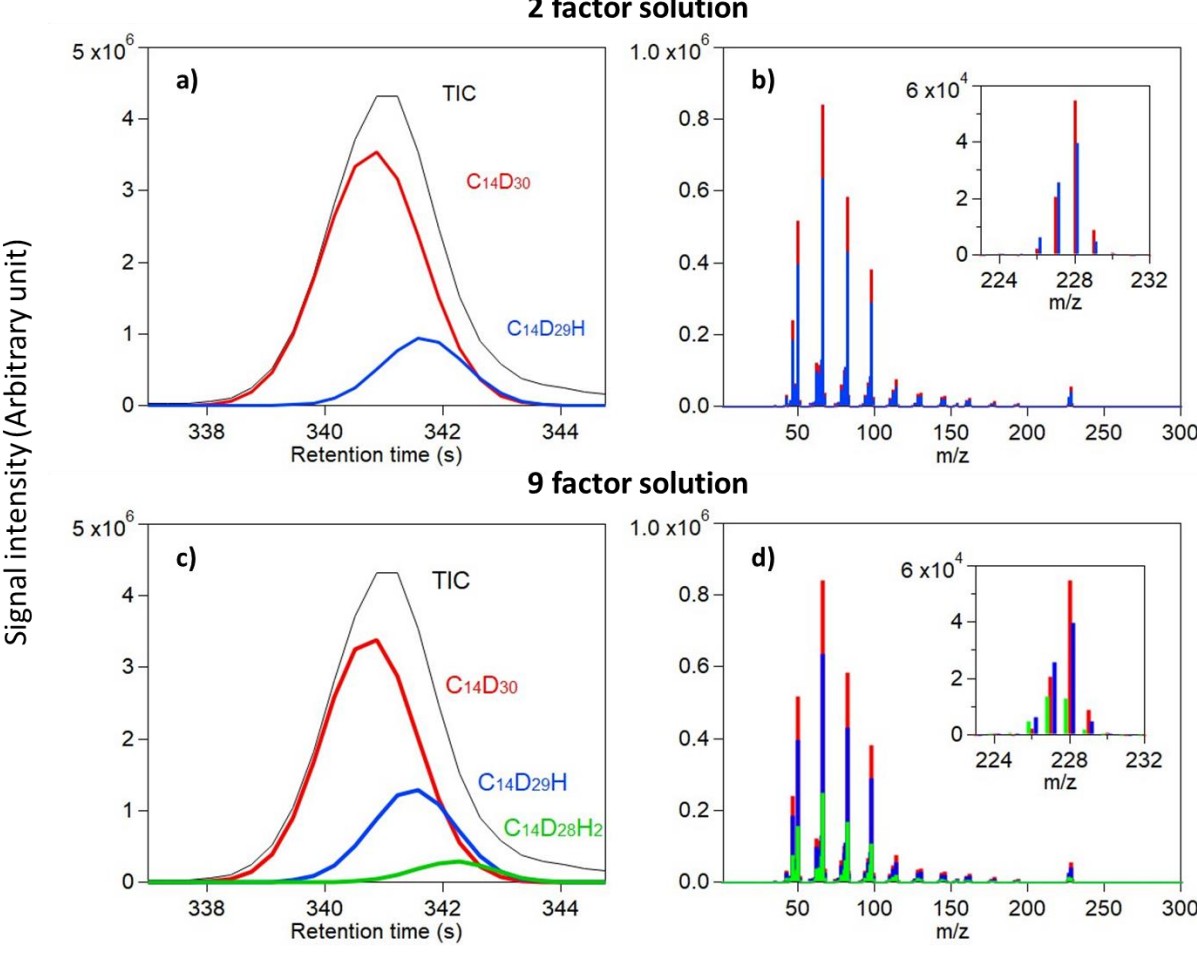

**Figure 4: Analysis of deuterated tetradecane. Total ion chromatogram and the reconstructed chromatographic profile of each analyte found in a (a) 2-factor solution and (c) 9-factor solution are displayed. Measured mass spectra at the corresponding location of each analyte found in a (b) 2-factor solution and (d) 9-factor solution are overlaid and displayed with a magnified view of their molecular weight peaks.**

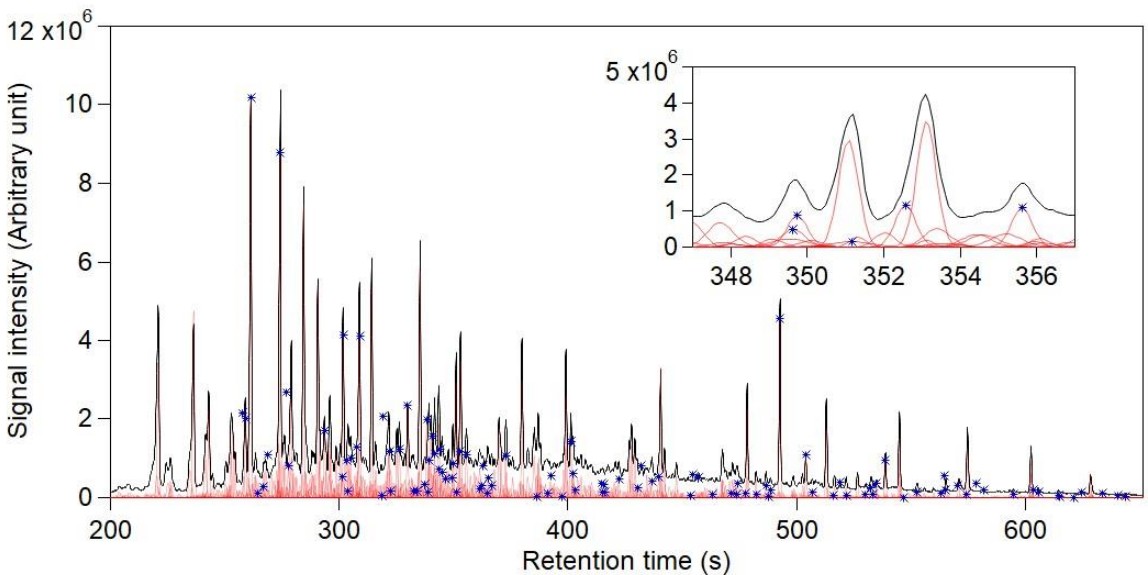

**Figure 5: Analysis of GoAmazon data using 4 chromatograms across the full 450-second chromatogram (200 – 650s). 10-second slices were used with 2-second overlap. Total ion chromatogram and reconstructed signal of each analyte are displayed in black and light red, respectively. The magnified view of a 10-second window (347-457s) is presented at the top right corner to display successful separation of co-eluted peaks. Blue stars indicate the analytes that were identified in a previously published manual analysis of this data.**
