# Peer review of "Comprehensive detection of analytes in large chromatographic datasets by coupling factor analysis with a decision tree"

_Atmospheric Measurement Techniques, 2022_

## Referee Comment (RC1)

Kim et al. developed a method to catalog analytes in chromatograms of complex environmental data. It was demonstrated that this method can identify more individual analytes information (~1000 species) than operators' manual inspection (~100 species) and save time from human-instrument interactions. Here I have some major comments.

1. Although the method developed by the authors can identify more than 10 times of analytes than the conventional one (manual inspection), what scientific problems can this method help us to solve? It seems that we still do not know most of the identified analytes.

I think the limit of identified analytes in previous studies is primarily due to the lack of authentic standards. The other big problem in investigating organic aerosols is the lack of organic tracers specifically related to their sources and/or transformation.

If the total ion chromatographs of environmental data were inspected manually based on individual $m/z$ ratios, at least several hundred of analytes will be identified, but mostly unknown.

Here, I have to admit that the method developed by the authors really simplified the time-consuming work for chromatograph inspection. But scientific problems should be solved to enlighten the significance of this work.

2. To keep column conditions, the GC column close to the inlet might need to be cut for quite a few centimeters after a batch of sample analysis, and retention times of analytes will vary differently. In this case, will the method be able to match the same analytes from different chromatograms in different batches?

3. How does the developed method deal with background peaks like pollution, column blood, etc? How does the method perform field blank correction?

---

## Referee Comment (RC2)

Review Kim et al., AMTD 2022

"Comprehensive detection of analytes in large chromatographic datasets by coupling factor analysis with a decision tree"

Kim et al. combined positive matrix factorization (PMF) and a decision tree for comprehensive peak detection in GC-MS datasets. Therefore, chromatograms are sliced in short sections and within these sections a certain (variable) number of factors is predetermined. These factors then contain a chromatographic profile along with its (fragmentation) mass spectrum. A decision tree algorithm discards factors that represent no compound. The combined PMF/decision tree data evaluation tool is successfully tested on a chromatogram of poorly separated deuterated tetradecane as well as on a complex ambient GC-MS dataset of the GreenOcean-Amazon field campaign.

Overall, the paper is well written, precise and comprehendible, even for non-PMF experts. The quality of the graphs is good and I can recommend the paper being published in AMT after addressing the following minor comments:

Minor comments:

l. 17: Why is "peak width" not included in peak evaluation of the decision tree?

l. 23 & l. 370: It is mentioned that 90% of the ~1100 "analytes" have no match with the NIST-database. Therefore, I suggest to use the term "features" instead of "analytes". This applies to the whole manuscript.

l. 30-35 or somewhere else in the introduction: Might be worth mentioning 2D-GC approaches to resolve complex samples.

l. 170 ff.: Peak-shapes of chromatograms are important and mostly not a perfect Gaussian. Peak shapes depend on the properties of the compound, but also on the condition of the column. That means, when analysing a few hundred of samples, the peak shape for one compound can become worse over time. How does the algorithm deal with that?
The authors used a Gaussian for chromatographic peak fitting and mention that a more complex approach is not necessary, although possible to include modified peak shapes. I disagree with the statement that non-Gaussian peak models are not necessary for proper peak fitting. An exponentially modified Gaussian (EMG) actually allows evaluating the Gaussian shape of chromatographic peaks by fitting with four variables (area, elution time, peak width and exponential) instead of just area, elution time and peak width (e.g. see Goodman & Brenna 1994). The mentioned paper by Isaacman-Van Wertz et al. (2017) is missing in the references.
How robust is the finding of the three different compounds (C14D30, C14D29H, C14D28H2) of the nine-factor solution when a non-Gaussian peak shape model is employed that allows to fit peak tailing? Since the difference is in the mass spectrum at *m/z* 226-230, I assume it is robust, but I can be wrong.

l. 174: The authors should provide evidence that "a refined peak shape is likely unnecessary", or otherwise should argue more carefully.

l.227: "M1 and M2 are normalized mass spectra of two analytes". This is confusing, because if the result is that epsilon>=0.8, then it is two normalized mass spectra of one analyte. I suggest rephrasing "M1 and M2 are normalized mass spectra selected for comparison".

Figure 5 shows several low-abundant compounds that were detected manually (blue asterisks). Were these compounds also detected by the presented PMF method? Why are the large prominent peaks

in the TIC not detected by the manual method? It looks if there is a homologue series of alkanes in the chromatogram (visible as an evenly-spaced series of decreasing peaks from 400-650 s). Why has this not been identified in the manual analysis? As a consequence, the manual inspection could easily identify much more compounds, with implication on the statement of the "one order of magnitude" (line 381).

Technical notes:

l. 30: HüBschmann → Hübschmann.

l. 249-250: use a non-breaking space between number and unit.

Literature:

Goodman KJ, Brenna JT (1994) Curve fitting for restoration of accuracy for overlapping peaks in gas chromatography/combustion isotope ratio mass spectrometry. Anal. Chem. 66(8):1294–1301.

---

## Author Comment (AC2)

Supplementary Information: Comprehensive detection of analytes in large chromatographic dataset using a coupled factor analysis/decision tree technique

Sungwoo Kim[1], Brian M. Lerner[2], Donna T. Sueper[2], Gabriel Isaacman-VanWertz[1,*]

[1]Charles E. Via Jr. Department of Civil and Environmental Engineering, Virginia Tech, Blacksburg, VA, 24061

[2]Aerodyne Research, Inc., Billerica, MA, 01821

S1. Example desciption of analysis process

The process of analyzing a 5-factor solution of a 10-second slice (335-345s) of an injection of known standards is presented here as an example. Four chromatograms of an alkane mixture sample are used, with one shown in Figure S1.

[Figure]

**Figure S1. Chromatogram of an alkane and deuterated alkane mixture. A 10-second subsection (335-345s) selected for the purpose of demonstration is displayed in red.**

In order to maximize our ability to determine whether peaks in the same slice of chromatograms are the same, a retention time refinement, modelled after correlation optimized warping (COW), is implemented. COW is a piecewise data preprocessing method that aligns the time profile of a sample towards a reference time profile by stretching or compressing the sample. The method used in this study, mode-ion COW, calculates the number of data points the single ion count (SIC) profile of the sample needs to be shifted by to maximize the correlation between the SIC profiles of the sample and reference. After calculating this parameter for all SICs within a slice, the most frequent number is used to shift the total ion count (TIC) of corresponding slice and chromatogram, and this step is repeated until all slices in all chromatograms are covered.

[Figure]

**Figure S2. Time profiles of 5 factors used in PMF analysis. Peaks corresponding to each analyte are numbered from 0-3.**

Four chromatograms are concatenated with an arbitrarily assigned time gap in between each chromatogram, with the chromatographic profiles of the resulting factors shown in Figure S2. Peak detection finds 16 peaks (numbered) across all 5 factors. Peak fitting results of Factor 3 are presented in Figure S3. Eight peaks are initially identified; these will eventually be sorted into two different analytes by the decision tree. From all four chromatograms, 16 Gaussian peaks out of 29 are sorted into four analytes (Figure S4) with known mass spectra (Figure S5).

[Figure]

**Figure S3. Gaussian curve fitting of factor 3. Fitted curves are numbered from 0-7 in increasing retention time order. The signal difference between the obtained time profile of factor 3 and the curve trace (the total added Gaussian signals), is labelled as fit residual and displayed on the top.**

[Figure]

**Figure S4. Gaussian curves of all analytes identified overlaid with the original chromatographic signal. The dashed line represents the total reconstructed signal of all analytes.**

[Figure]

**Figure S5. Mass spectra of 4 analytes identified.**

S2. Analysis results of a subsection containing tetradecane (alkane mixture sample)

[Figure]

**Figure S6. Analysis results of tetradecane, analogous to Figure 4). TIC and the reconstructed time profile of each analyte found in a (a) 2-factor solution and (c) 9-factor solution are displayed. Measured mass spectra at the location of each analyte found in a (b) 2-factor solution and (d) 9-factor solution are stacked and displayed with a magnified view of their molecular weight peaks.**

[Figure]

**Figure S7.** Analysis results of deuterated tetradecane with an exponentially modified gaussian (EMG) as a peak fitting model. EMG curves of all analytes identified overlaid with the original chromatographic signal. The purple dashed line represents the total reconstructed signal of all analytes. Red cross markers represent the location and height of each corresponding analyte. The retention time difference between $C_{14}D_{30}$ and $C_{14}D_{29}H$ is labelled as 'a' and the latter as 'b'.

An experiment has been conducted to investigate the impact of employment of exponentially modified gaussian (EMG) as a peak fitting model, using this same subsection as an example. The analysis method was applied with nine factors on a 15-second chromatographic window of known liquid standards samples described in section 2.3. Since the notable difference in the mass spectra is only detected at the molecular mass ions, their cosine similarity values are above the predetermined threshold ($\varepsilon > 0.8$). However, the median half width at half maximum (HWHM), is smaller than the separation between peaks, but more conversative thresholds for the critical retention time difference would combine some or all of these peaks. The calculated values of median HWHM, a, and b are 0.612, 0.825, and 0.677, respectively.

S3.  Results of a single slice of GoAmazon data.

[Figure]

**Figure S8. Number of analytes identified using various subsection sizes and a range of factors (2-35).**

The analysis results of a known alkane mixture clearly show the rate of new information acquired decreases as the number of factors increases. This indicates that a balance between the level of information and computational time can be found. With this knowledge, multiple analyses were performed on a randomly selected slice of the real-world test data to determine a preferred subsection size and number of factors to be used for the analysis of the entire dataset. Number of analytes found in various subsection sizes (5,6,8, and 10s) were recorded as the number of factors were ranged from 2 to 35. An approximate ratio of roughly 2.5 factors needed per second of time in the slice was determined as the approximate level beyond which increasing factors does not substantially increase the amount of information extracted. Additional factors may provide some additional information but would substantially increase computational time (Figures S9 and S10); the balance between comprehensive analysis and computational time inherently needs to be weighed based on the preferences and available resources of the user.

[Figure]

**Figure S9. Computation time of one slice using various subsection sizes and a range of factors (2-35).**

[Figure]

**Figure S10. Calculated computation time for analysis of the entire retention time range (200-650 s) using various subsection sizes and a range of factors (2-35).**

[Figure]

**Figure S11. Distribution of mass spectra forward matching values compared to the NIST MS library.**

The mass spectra of all 1169 analytes were compared to the NIST MS library using forward matching metric. NIST guideline describes a match value ranging from 800 to 900 as a "good match" and from 900 to 1000 as an "excellent match". A total of 96 analytes were confirmed with match values above 800.

**Table S1**. **Number of analytes cataloged by using various critical retention time difference values.**

| Name | Definition | Retention time (s) | Number of analytes |
|---|---|---|---|
| HWHM | $1.177\sigma$ | 0.328 | 1216 |
| MPF* | $\sqrt{2}\sigma$ | 0.394 | 1169 |
| 4 datapoints | $1.5\sigma$ | 0.420 | 1145 |
| $2\sigma$ | $2\sigma$ | 0.558 | 1018 |
| FWHM | $2.355\sigma$ | 0.656 | 943 |

*Used by the Multi-peak fitting 2 package used for analysis to simplify calculations*

The impact of screening potential analytes with various critical retention time difference values has been examined. The ambient aerosol samples described in Section 2.3 were analyzed using the method described in Section 3.3 with several critical retention time differences. The results show that the number of analytes cataloged decreases as a more conservative approach is used. As compared to the analysis results presented in Fig. 5, approximately 20 percent fewer analytes were cataloged by using the most conservative (approximately 66 percent larger in retention time) critical retention time difference. Since all other conditions for analysis are kept identical throughout the experiment except for the critical widths, the reduction in the number of cataloged analytes is independent of their mass spectra.

---

## Author Response (AR1)

We thank the reviewers for their careful consideration and recognition of the value of this work. Please find below our responses to all comments and associated changes to the manuscript. Original comments are included in **bold**, and changes to the manuscript are excerpted in *italics*.

Reviewer 1

**1. Although the method developed by the authors can identify more than 10 times of analytes than the conventional one (manual inspection), what scientific problems can this method help us to solve? It seems that we still do not know most of the identified analytes. I think the limit of identified analytes in previous studies is primarily due to the lack of authentic standards. The other big problem in investigating organic aerosols is the lack of organic tracers specifically related to their sources and/or transformation. If the total ion chromatographs of environmental data were inspected manually based on individual m/z ratios, at least several hundred of analytes will be identified, but mostly unknown. Here, I have to admit that the method developed by the authors really simplified the time-consuming work for chromatograph inspection. But scientific problems should be solved to enlighten the significance of this work.**

We agree that a lot of work remains to be done to identify the unknown compounds in the atmosphere, and indeed agree that the limit is in many cases the availability of authentic standards. We note that many tracers now commonly used by the community started out as components with unknown structure or origin (for example, $C_5$ alkene triols that are commonly measured as isoprene oxidation tracers required significant dedicated effort to identify (Wang et al. 2005)). Recently, a focus of the atmospheric community of GC/MS users has been to build up libraries of these types of compounds and works show that unknowns can be useful tracers (e.g., multiple libraries available through Dr. Allen Goldstein at UC Berkeley: https://nature.berkeley.edu/ahg/resources/). We also highlight previous work that used correlation to known tracers to identify the likely sources of unknown compounds (e.g., Isaacman-VanWertz, G. et al. 2016), and in some cases used this information to quantitatively attribute sources of aerosol (Zhang, H. et al. 2018). Even in the lack of current identification, we therefore believe it is useful to integrate and investigate all analytes, even those that are unknown at the moment, and examine the data as a whole to determine if any minor components may serve as useful tracers or provide unique information.

Further, we agree that manual inspection can capture many of the analytes in a dataset as proposed, and a central goal of the current work is to increase the efficiency of this process. However, there are also advantages to the proposed process relative to manual inspection other than simple decreases in processing time (which is nevertheless a major advance of this work). Perhaps most importantly, the current work outputs not only the location of an analyte, but also a "clean" mass spectrum that can be used comparison to known libraries (and identification, though that is a rare case as discussed). By using multiple chromatograms to extract this spectral information, this output is more robust that could be achieved through manual inspection. Furthermore, by including multiple chromatograms in the analysis, components that are present in only one or one type of chromatogram are also cataloged, which might be overlooked in manual inspection (which is not practical to do thoroughly for many chromatograms). Lastly, as illustrated in Fig. 4, this method is able to resolve peaks that would otherwise be nearly impossible under manual inspection. This is useful in identifying low signal analytes which can yield new insight into sources and transformations of organic compounds in the samples.

For the reasons described here, we believe the method has significant value to address research questions, ranging of quantifying aerosol sources to a more complete understanding of the impacts of atmospheric processes on atmospheric composition. However, given the complexity of the analytical approach, we believe it is best to focus this manuscript on the approach itself and allow future work to focus on the scientific advances enabled.

The following sentences have been added to clarify the issue.

*"Significant work remains to be done to identify the unknown compounds in the atmosphere, however many tracers commonly used by the community started out as components with unknown structure or origin. For example, C5 alkene triols that are commonly measured as isoprene oxidation tracers required significant dedicated effort to identify (Wang et al. 2005). Previous work has also been done wherein correlation to known tracers was used to identify the likely sources of unknown compounds (Isaacman-VanWertz, G. et al. 2016), and in some cases, this information was used to quantitatively attribute sources of aerosol (Zhang, H. et al. 2018). Therefore, despite the lack of current identification, we believe it is useful to integrate and investigate all analyte, and examine the data as a whole."*

Wang, W. et al. Characterization of oxygenated derivatives of isoprene related to 2-methyltetrols in Amazonian aerosols using trimethylsilylation and gas chromatography/ion trap mass spectrometry. Rapid Commun. Mass Spectrom. 19, 1343–1351 (2005).
Isaacman-VanWertz, G. et al. Ambient Gas-Particle Partitioning of Tracers for Biogenic Oxidation. Environ. Sci. Technol. 50, 9952–9962 (2016).
Zhang, H. et al. Monoterpenes are the largest source of summertime organic aerosol in the southeastern United States. Proc. Natl. Acad. Sci. U. S. A. 115, 2038–2043 (2018).

**2. To keep column conditions, the GC column close to the inlet might need to be cut for quite a few centimeters after a batch of sample analysis, and retention times of analytes will vary differently. In this case, will the method be able to match the same analytes from different chromatograms in different batches?**

As the reviewer notes, aligning chromatographic signals is necessary before applying statistical data reduction methods such as principal component analysis (PCA) and positive matrix factorization (PMF) (Eilers 2004). In the present work, a coarse retention time adjustment was applied manually, with a subsequent fine-scale retention time adjustment that was conducted as part of the automated cataloging approach. The current implementation of this process does not include the initial global adjustment, which we agree would be necessary for application to chromatograms from different batches.

Others have described relatively robust global retention time alignment methods that could be automated (e.g., parametric time warping, Eilers 2004), and we are actively working to integrate such an approach into the peak cataloging software as a pre-processing step to make the entire process more automated. However, doing so would be primarily an implementation of previously demonstrated methods, so a detailed discussion is not included in this manuscript. Instead, we have revised the discussion on line 131 to make this issue clear.

*"Each chromatogram is first aligned to the same retention time basis by using a small number of known compounds or introduced standards in each sample to define known retention times. Strictly speaking,*

*this preprocessing is not necessary for factor analysis. However, interpretation of the outcome of data reduction techniques such as PARAFAC(2) and PMF can be unreliable when chromatograms are used directly as input (Eilers, 2004; Van Nederkassel et al., 2006), as it may be difficult or impossible to determine if unaligned peaks in each chromatogram represent the same analyte. Chromatogram alignment may occur through manual adjustment by users or may be automated using any of multiple solutions (Eilers, 2004; Kassidas et al., 1998; Nielsen et al., 1998), but the cataloging approach described here is independent of the details of any such approach (a manual approach is used in this work), so details are not included."*

**3. How does the developed method deal with background peaks like pollution, column blood, etc? How does the method perform field blank correction?**

Multiple samples are analyzed simultaneously using this method and positive matrix factorization (PMF) extracts a commonly observed pattern from all the samples. With the assumption that a high enough number of factors were used for the analysis, any existing analytes with a significant level of signal should be identified as separate analytes by this method, regardless of whether such an analyte is a compound present in the sampled air or is a contaminant. It is up to the user on how to utilize the outcome analyte information. In some analyses our approach has been to include a chromatogram from a background or blank to identify components that are present in these background samples as well.

The manuscript has been revised to clarify the issue.

*"In contrast to other PMF applications, the primary goal in this work is not to optimally describe the complete data set, but rather to increase the number of factors to a point where even minor components are extracted as separate factors, even at the risk of over-fitting the data (which will be rectified by a subsequent decision tree). With this approach, any existing analytes with a significant level of signal should be identified as separate analytes, regardless of whether such an analyte is a compound present in the sample or is a contaminant."*

Reviewer 2: Alexander Vogel

**l. 17: Why is "peak width" not included in peak evaluation of the decision tree?**

As originally written, this sentence was not clear, using the phrase "peak shape" to represent not only the distinctive pattern of time profile (e.g., Gaussian or Exponentially modified gaussian) but also the peak width, height, and their relative ratios. Line 17 has been revised as *"A decision tree based on peak parameters (e.g., location, width, and height), relative ratios of those parameters, noise, retention time, and mass spectrum is applied to…."*

**l. 23 & l. 370: It is mentioned that 90% of the ~1100 "analytes" have no match with the NISTdatabase. Therefore, I suggest to use the term "features" instead of "analytes". This applies to the whole manuscript.**

We acknowledge that the word 'analyte' is often used to refer to identified chemical substances, and we appreciate the suggestion to adjust the wording so as to avoid potential confusion for the readers. However, within this community, the term "analyte" has been frequently used to refer to a feature in a sample whether or not it has a known definitive identification (e.g., references from the manuscript: Amigo et al 2010, Isaacman-VanWertz et al 2017, and others in this journal: Li et al., 2022 and Grace et al., 2019). We are of the opinion that the term "analyte" will be less confusing to the broader atmospheric community and readership of this journal. To avoid any confusion, we have clarified our definition in the manuscript as excerpted below. We leave the final decision on this choice to the editorial staff of the journal and are open to making the switch from "analyte" to "feature" if the editor or journal staff feels it would be better suited.

*"In this work, the term "analyte" is used to refer to a unique chromatographic peak (e.g., a chromatographic "feature") whether it has a known definitive identification or not, following the usage of this term other studies (Amigo et al., 2010; Grace et al., 2019; Isaacman-Vanwertz et al., 2017; Li et al., 2022)."*

Li, H. et al.: Fragmentation inside proton-transfer-reaction-based mass spectrometers limits the detection of ROOR and ROOH peroxides, Atmos. Meas. Tech., 15, 1811–1827, https://doi.org/10.5194/amt-15-1811-2022, 2022.
Grace, D. N. et al.: Separation and detection of aqueous atmospheric aerosol mimics using supercritical fluid chromatography–mass spectrometry, Atmos. Meas. Tech., 12, 3841–3851, https://doi.org/10.5194/amt-12-3841-2019, 2019.

**l. 30-35 or somewhere else in the introduction: Might be worth mentioning 2D-GC approaches to resolve complex samples.**

We agree that GCxGC holds promise for tackling some of these issues and warrants inclusion in the introduction. The following sentences have been added to the manuscript.

*"The resolution of GC can be expanded by coupling multiple columns in series, and comprehensive two-dimensional gas chromatography (GC×GC) can provide greater sensitivity and resolution of complex mixtures (Bertsch, 1999; Phillips and Beens, 1999). This technique has yielded valuable insights into atmospheric composition (Hamilton, 2010), but the increased complexity of the instrumentation and more stringent requirements for the mass spectrometer (e.g., time resolution faster than ~50 Hz (Worton*

*et al., 2012)) has limited adoption of GC×GC. Furthermore, despite the higher resolving power, co-elution of peaks still occurs (Potgieter et al., 2016) when highly complex samples are analyzed, and challenges remain in the data analysis. Therefore, it is consequently common for analyses of environmental data to focus on the resolution and quantification of only a subset of specific analytes of interest and leave a large fraction of data unprocessed and unused."*

**l. 170 ff.: Peak-shapes of chromatograms are important and mostly not a perfect Gaussian. Peak shapes depend on the properties of the compound, but also on the condition of the column. That means, when analysing a few hundred of samples, the peak shape for one compound can become worse over time. How does the algorithm deal with that?**

As described in lines 101-110, positive matrix factorization (PMF) extracts a commonly observed pattern from the dataset as a factor which consists of a chromatographic signal and corresponding mass spectrum. The time profile of a factor is a true reflection of the peak shape (Gaussian or otherwise) in each chromatogram. A changing peak shape over time will be reflected in the profiles. Fitting to a Gaussian (or other shape) peak is used simply to catalog or describe the peak in each chromatogram. As peak shape changes from sample to sample, the parameters in the fit would represent that, and if a deteriorating shape begins to deviate from Gaussian, the width of the fit may become less representative of the true width. However, the location, mass spectrum, and approximate height of the peak should be reasonably well documented in the catalog, enabling a user to find and integrate the peak using whatever peak shape best describes the data.

**The authors used a Gaussian for chromatographic peak fitting and mention that a more complex approach is not necessary, although possible to include modified peak shapes. I disagree with the statement that non-Gaussian peak models are not necessary for proper peak fitting. An exponentially modified Gaussian (EMG) actually allows evaluating the Gaussian shape of chromatographic peaks by fitting with four variables (area, elution time, peak width and exponential) instead of just area, elution time and peak width (e.g. see Goodman & Brenna 1994). The mentioned paper by IsaacmanVan Wertz et al. (2017) is missing in the references.**

Incorporating an EMG model for peak fitting increases the ability of a fitting algorithm to describe real world chromatographic peaks, though with increased possibility of poor fitting of poorly resolved peaks due to the additional degrees of freedom. In this study, the primary goal is catalog peaks, with peak fitting used primarily to extract the parameters to describe the peak (location, height, width); indeed, peak detection is agnostic toward peak shape, relying on first and second derivatives to identify inflection points in the factor profiles. Consequently, we apply only a Gaussian model to lower the level of complexity in the calculation of the parameters describing each peak. However, we agree that an EMG peak shape may improve the accuracy of these parameters (particularly peak width), and have clarified our reasoning and language in the lines 191-197 and 460-464 of the revised manuscript:

"A more complex approach could include modified peak shapes (e.g., convolution with an exponential (Isaacman-VanWertz et al., 2017)), *which would likely enable more accurate characterization of the parameters describing a peak. However, in this work the goal is to catalog all peaks by their approximate parameters as opposed to perfectly integrate them, so increasing the complexity of peak fitting by incorporating refined peak shapes has not been implemented. Implementation of exponentially modified gaussian (EMG) as a peak fitting model has been inspected on the samples containing deuterated*

*tetradecane presented in Fig.4 and discussed in the supplementary information (Fig. S7). Optimal peak shapes could be used in subsequent processing for accurate integration of data."*

*"Three major future developments would further enhance this approach: improved retention time correction without human interaction (e.g., by parametric optimized warping (Eilers, 2004)), incorporation of modified peak shapes (e.g., exponentially modified gaussian) as a peak fitting model, and algorithmic optimization of decisions around the length of each slice, the number of factors, and the number of chromatograms."*

A far more expansive investigation of EMG peak shapes has also now been included as discussed in response to the comment below.

**How robust is the finding of the three different compounds (C14D30, C14D29H, C14D28H2) of the nine-factor solution when a non-Gaussian peak shape model is employed that allows to fit peak tailing? Since the difference is in the mass spectrum at m/z 226-230, I assume it is robust, but I can be wrong.**

We would like to thank the reviewer for raising this discussion, which prompted us to examine more deeply decisions around the retention time difference parameter in the decision tree. We have added a discussion of the reviewer's specific comment and significantly revised discussion surrounding peak shape and width. As described in lines 242-246, the retention time difference and cosine similarity value are the two main criteria used in the separation of peaks in the decision tree. Prompted by this comment, we have considered the retention time difference more deeply; while it is inherently a function of peak width, there is also some flexibility and implications in how the critical retention time difference should be set so we have switched from calling this parameter peak width to "critical retention time difference" and discuss it in more detail.

Though there exists some difference in the mass spectra of the compounds $C_{14}D_{30}$, $C_{14}D_{29}H$, and $C_{14}D_{28}H_2$, but due to heavy fragmentation of alkanes, the mass spectra are sufficiently similar ($\varepsilon > 0.8$) that it is difficult to algorithmically separate with cosine similarity as a sole metric. These peaks are instead being separated primarily by retention time difference. Considering that the differences in the high molecular weight ion distributions, and the predictable rightward shift of the retention times of each isotopologue, we are reasonably confident that these are unique analytes and thus present an opportunity to examine the impacts of how the retention time difference criterion is selected and the potential impact of peak shape. We find, overall, that there is a strong argument for allowing some control by the user over that argument and examine the impacts of this control, which balances a tendency for negative and positive errors.

In the discussion of peak sorting, in which peaks in the same factor are separated into analytes with same mass spectrum, we have added:

*"Selection of a critical retention time difference is somewhat dependent on the goals of the user but is inherently related to peak widths. A conservative estimate of a critical width is several times the standard deviation (e.g., FWHM = 2.355σ), which would ensure that only peaks that are truly chromatographically resolved are regarded as unique. However, in many cases, isomers may not be well resolved but nevertheless represent unique analytes, which may be apparent in small changes in ion ratios or signal intensities across chromatograms. In these cases, a more aggressive (i.e., smaller)*

*approach to critical retention time differences may be appropriate, which might include HWHM (~1.18 σ), or, most aggressively, peaks that are separated by only one or two datapoints (i.e., a peak in a different time period of instrument acquisition). Setting this parameter more aggressively increases the possibility of positive errors, discussed in Section 2.4."*

And we note that xylenes are an example of this case:

*"Isomers such as these represent an example of the potential impact of a user-specified critical retention time difference, as an aggressive value (e.g., one or two datapoints) may separate these analytes if there is at least some separation by retention time and some variability in ratios between samples that may be detected by the PMF, while a more conservative approach (e.g., FWHM) is unlikely to separate poorly resolved isomers."*

We further discuss the selection of the critical retention time difference in the analyte sorting discussion when peaks in separate factors are combined. We note that in this case, there is some measurable difference in spectra and/or sample-to-sample variability, since peaks have been separated by PMF even if cosine similarities are not strictly above the threshold:

*"Again, the selection of the critical retention time difference exerts some control on the opposing tendencies of this approach to either consider peaks unique (potentially leaving multiple peaks representing the same analyte) or combine peaks (potentially binning multiple analytes). In this step, any potential analytes being compared must exhibit at least some difference in mass spectrum and sample variability, since they were separated by the PMF, so a more aggressive critical retention time difference is likely warranted here."*

With this in mind, we have specified our critical retention time differences throughout the manuscript. Specifically, we note that in our analysis of analytes used for Fig. 3 we use a relatively aggressive value:

*"The critical retention time difference used in this analysis was relatively aggressive (median HWHM, which equals 0.7s in this data) in order to examine the capability of the method to find unique peaks; the effects of this selection are discussion below."*

An extensive discussion of the effect of retention time difference on isotopologue separation has also been added in direct response to this comment:

*"Separation of these isotopologues presents an opportunity to examine the impact of the critical retention time difference, and the impact of assumed Gaussian peak shapes on this separation. Though exhibiting interpretable differences in their higher molecular weight ions, the heavy fragmentation of alkanes yields mass spectra that are not sufficiently different to be separated by the cosine similarity threshold (i.e., comparisons between all three isotopologues have $\epsilon \geq 0.8$), despite sufficient differences to be separated into different factors in the PMF. Consequently, resolution of these peaks relies on separation by retention time in the analyte sorting step. Separation between each peak is roughly 0.75 second in retention time, while median peak width in the dataset (σ) is 0.6s, peak widths of these analytes are roughly on the order of 0.7s, and a mass spectrum is collected every 0.3s. Peaks are consequently separated by more than two datapoints, and more than the median HWHM of the dataset (0.71s), but not by more than the HWHM of these specific peaks (0.82s) or by more than the median FWHM of the dataset (1.4s). In other words, only more aggressive screening methods (i.e., using σ or median HWHM as the critical retention time difference) would separate these isotopologues. This*

*approach also increases the chance of chromatographic artifacts being cataloged as real analytes (positive error), but a more conservative approach increases the possibility of overlooking poorly resolved and similar analytes such as these (negative error). Ultimately, it is up to the user to decide the optimal critical retention time difference."*

The effect of a modified peak shape is found to be relatively minor. Because PMF is agnostic to peak shape and so is peak detection (which uses first and second derivatives), the impact of EMG fitting is only on the widths of the peak and the potential for combining them in peak sorting and analyte sorting. So the issue is again one of setting the critical retention time threshold. We find the results to be essentially the same as those achieved using Gaussian fitting, that more aggressive values separate the isotopologues while less aggressive values do not. That has been included in the revised manuscript as well as an associated figure and discussion in the SI:

"*The effect of a non-Gaussian peak shape was also examined. Because peak detection relies on derivatives to identify potential peaks based on inflection points in the data, the number of peaks found is agnostic toward peak shape; instead, peak shape primarily impacts peak widths. Using an exponentially modified Gaussian peak shape to the analysis of isotopologues does not substantially change the result (Fig. S7). With this peak shape, isotopologues remain separated using more aggressive critical retention time differences (median FWHM or more than two datapoints) but are combined by more conservative thresholds. This result is of course limited to the shown case, in which a Gaussian curve reasonably describes the observed data. Datasets containing highly non-Gaussian peak shapes may be more impacted and should be examined closely for the potential impact of peak tailing on positive errors*."

[Figure]

**Figure S7. Analysis results of deuterated tetradecane with an exponentially modified gaussian (EMG) as a peak fitting model. EMG curves of all analytes identified overlaid with the original chromatographic signal. The purple dashed line represents the total reconstructed signal of all analytes. Red cross markers represent the location and height of each corresponding analyte. The retention time difference between $C_{14}D_{30}$ and $C_{14}D_{29}H$ is labelled as 'a' and the latter as 'b'.**

With an EMG model, the compounds are separated since the median HWHM is smaller than both a and b. However, it is notable that the HWHM of $C_{14}D_{30}$ is greater than a. Ideally, each pair of co-eluting peaks is investigated, and a proper width needs to be determined for each pair and used in the decision tree. Programmatically, this would be challenging and thus a representative value, median HWHM, is compared to the retention time differences. Ultimately, it is up to the user to decide what definition of width should be used as an efficient screening tool.

**l. 174: The authors should provide evidence that "a refined peak shape is likely unnecessary", or otherwise should argue more carefully.**

As described in response to the reviewer's comments above, lines 191-195 have been revised:

"A more complex approach could include modified peak shapes (e.g., convolution with an exponential (Isaacman-VanWertz et al., 2017))*, which would likely enable more accurate characterization of the parameters describing a peak. However, in this work the goal is to catalog all peaks by their approximate parameters as opposed to perfectly integrate them, so increasing the complexity of peak fitting by incorporating refined peak shapes has not been implemented. Optimal peak shapes could be used in subsequent processing for accurate integration of data."*

**l.227: "M1 and M2 are normalized mass spectra of two analytes". This is confusing, because if the result is that epsilon>=0.8, then it is two normalized mass spectra of one analyte. I suggest rephrasing "M1 and M2 are normalized mass spectra selected for comparison".**

We agree this can be confusing. The phrase 'potential analytes' have been used throughout the manuscript to refer to these undetermined substances. We have clarified using a combination of this and the suggested language: *"M1 and M2 are normalized mass spectra of two potential analytes being compared to determine whether they represent the same analyte."*

**Figure 5 shows several low-abundant compounds that were detected manually (blue asterisks). Were these compounds also detected by the presented PMF method? Why are the large prominent peaks in the TIC not detected by the manual method? It looks if there is a homologue series of alkanes in the chromatogram (visible as an evenly-spaced series of decreasing peaks from 400-650s). Why has this not been identified in the manual analysis? As a consequence, the manual inspection could easily identify much more compounds, with implication on the statement of the "one order of magnitude" (line 381).**

The reviewer makes a good point, that as described in the original manuscript, the previously published analysis of these data "focused on only ~100 compounds," not that those were the only analytes that could be possibly identified in this dataset. Indeed, the alkanes could be identified, though there were not a focus of that analysis or present in the published work (which focused on oxidation products of biogenic emissions). We examine this dataset in this work with two goals in mind: to test for negative artifacts (overlooking analytes known to exist in the dataset) and to determine the number of analytes cataloged as a representative application. We have clarified these goals at the start of section 3.3:

"To evaluate the proposed method in a real-world application, we apply it across the full chromatographic range for data representing the gas- and particle-phase composition of atmospheric samples. *The goal of this analysis is to both provide an estimate of the number of analytes found in representative atmospheric samples and evaluate the ability of the cataloging approach to identify analytes known to exist in a complex, real-world dataset."*

We have further clarified that the 100 compounds focused on may not represent all manually identifiable compounds, on line 430, and note the additional benefits beyond simply finding a higher number of analytes:

"In contrast, a previously published analysis of this dataset focused on only ~100 compounds cataloged by manual inspection, *though additional compounds are observed to exist in the dataset that were not a focus on this previous analysis. We note that a major advantage of the proposed approach is not only the larger number of analytes cataloged (with significantly less manual interaction), but also that each of these analytes has a well-defined mass spectrum that can be used for identification or comparison to existing mass spectral libraries."*

We have also removed the language from the conclusion regarding "an order of magnitude" as we agree that is dependent on how many peaks manual inspection can find, which is a function of time and effort costs, and instead focus on the more quantifiable conclusions that "*more than 1000 analytes were cataloged* with little or no human interaction."

In addition, we have examined how our decision to set a critical retention time difference may impact these numbers. We find that aggressive thresholds increase the value to ~1200, and very conservative thresholds (FWHM) decrease it to 950, so roughly 1000 analytes appear to be a fairly robust conclusion. This has been included in the manuscript as well as a table of the results in the SI.

*"This analysis uses a moderately aggressive critical retention time difference (1.4 σ), but the number of analytes found is slightly reduced by more conservative approaches (e.g., only 20% lower at using the much more conservative FWHM, Table S1)."*

**Table S1. Number of analytes cataloged by using various critical retention time difference values.**

| Name | Definition | Retention time (s) | Number of analytes |
|---|---|---|---|
| HWHM | $1.177\sigma$ | 0.328 | 1216 |
| MPF* | $\sqrt{2}\sigma$ | 0.394 | 1169 |
| 4 datapoints | $1.5\sigma$ | 0.420 | 1145 |
| $2\sigma$ | $2\sigma$ | 0.558 | 1018 |
| FWHM | $2.355\sigma$ | 0.656 | 943 |

**Technical notes:**

**l. 30: HüBschmann → Hübschmann.**

**l. 249-250: use a non-breaking space between number and unit.**

The manuscript has been revised accordingly.